# *SUPERhip* Reconstruction Treatment in Patients with Congenital Femoral Deficiency of Paley’s Classification Type 1b

**DOI:** 10.3390/children11010075

**Published:** 2024-01-09

**Authors:** Francisco Hélio Violante Júnior, Fernando Farcetta Júnior, Douglas Manuel Carrapeiro Prina, Monica Paschoal Nogueira

**Affiliations:** 1IAMSPE–Instituto de Assistência Médica ao Servidor Público Estadual, São Paulo 04029-000, SP, Brazil; franciscoviolante@uol.com.br (F.H.V.J.); carrapeiroprina@gmail.com (D.M.C.P.); 2Associação de Assistência à Criança Deficiente—AACD, São Paulo 04027-000, SP, Brazil; ffarcetta@aacd.org.br

**Keywords:** hip deformity, coxa vara, varus of the proximal femur, valgus osteotomy, congenital femoral deficiency

## Abstract

Background: Congenital femoral deficiency (CFD) is a challenging and complex condition that causes limb lengthening. We focused on the clinical and radiographic results of *SUPERhip* procedures in patients with congenital femoral deficiency type 1b, according to Paley’s classification, prior to femoral lengthening. Methods: We reviewed all records and radiographs of patients who underwent this procedure between 2005 and 2020. We included 26 patients, with clinical and radiographic assessments performed during pre- and post-operative evaluations. Results: There were twenty-six patients (15 right vs. 11 left), with a mean age of 7 years (1–18). Most of the patients were female (17 patients; 65.4%). Twenty (76.9%) patients were undergoing their first procedure and six (23.1%) had already undergone a previous surgery. There was a significant improvement in all radiographic parameters, with the mean preoperative and postoperative Neck-Shaft Angle (NSA) being 72.3 ± 7.1° vs. 133.1 ± 12.7°, the Center-Edge Angle (CEA) 16.8 ± 9.8° vs. 33.5 ± 14.1°, and the Acetabular Index (AI) 27.8 ± 6.9° vs. 16.4 ± 6.8°, respectively. The complication rate was 15.4%, predominantly affecting patients under 5 years old. Conclusions: The *SUPERhip* procedure is an effective and reproducible technique for clinical and radiographic correction to a significant degree in patients with Paley’s type 1b CFD, in preparation for bone lengthening surgery.

## 1. Introduction

Congenital femoral deficiency (CFD) is a complex condition that presents challenges to orthopaedic surgeons in limb reconstruction and lengthening surgery.

Generally, the femur is very short, but the term congenital femoral deficiency is broader and includes apparent coxa vara and abduction contracture of the hip (tensor fascia lata, gluteus medius, and gluteus minimus muscles), deformities of the proximal femur in extension, or even bony flexion of the neck in some cases, with concomitant flexion contracture of the hip involving the rectus femoris, psoas, and tensor fascia lata muscles, as well as retroversion of the femur with soft tissue contractures in external rotation. In addition, the proximal femur contains non-ossified regions (delayed ossification).

There can be valgus deformities caused by hypoplasia of the lateral femoral condyle. The total discrepancy at maturity can be estimated, even in the first years of life, by measuring the difference in length on a panoramic radiograph, with blocks placed under the short leg until both iliac crests appear to be level and multiplying this difference by the multiplier number for age, as is already described in the literature [1]. Although treatment with lengthening is important, it should not be started until the thigh and femur have been properly reconstructed. Hip instability during femoral lengthening can cause subluxation and dislocation, which is difficult to manage [2].

Treatment of Paley’s type 1a (normal ossification) and type 1b (delayed ossification) deformities has evolved greatly with knowledge of the anatomopathology of this orthopedic alteration, making it possible to lengthen the femur with fewer complications. The specific procedure for reconstructing the hip in patients with type 1b CFD is called *SUPERhip* (SUPER is an acronym for Systematic Utilitarian Procedure for Extremity Reconstruction, described by Paley) and involves multiple procedures such as lengthening the soft tissues (especially the flexor and abductor muscle–tendon structures of the hip), an unconventional valgus, extension, and internal rotation femoral osteotomy, and periacetabular osteotomy [3,4,5,6,7,8].

The aim of this study is to report the clinical and radiographic results of *SUPERhip* (SH) reconstruction surgery in patients with CFD type 1b according to Paley’s classification, prior to femoral lengthening. The target readership is pediatric orthopedists.

## 2. Materials and Methods

Institutional review board approval of “Instituto de Assistência Médica ao Servidor Público Estadual” (protocol code 16377119.4.0000.5463 and date of approval in 22 July 2019) was obtained for a retrospective review of all charts and radiographs of patients who underwent a *SUPERhip* procedure for CFD between October 2005 and March 2020.

This procedure is based on femoral osteotomy, for the correction of primary deformities of the proximal femur, and pelvic osteotomy for the improvement of acetabular coverage, described in detail below in Section 2.2.

The inclusion criteria were as follows: (1) patients diagnosed with unilateral CFD type 1b according to Paley’s classification; (2) Follow-up minimum of 3 years.

### 2.1. Patients

In a period of 15 years, the *SUPERhip* procedure was performed on 26 patients with a diagnosis of CFD type 1b, according to Paley’s Classification [4] at the initial consultation. There was delayed ossification of the femoral neck in 16 patients (61.5%), it was subtrochanteric in 5 cases (19.3%), and 1 patient (3.8%) had a combination of both subtypes.

At the time of surgery, it was noted that four patients had evolved to subtype 1a (older patients) due to spontaneous delayed femoral ossification. All the patients had unilateral involvement, confirmed by the author at the first consultation.

Informed consent was obtained from all subjects involved in the study.

### 2.2. Planning

*SUPERhip* determines the time of lengthening and the prognostics of each patient. We evaluated individuals and carried out the procedure as described by Paley [3]. Lengthening should be started after the upper femur has been adequately reconstructed.

Anteroposterior radiographs of the hip, bilateral hip abduction, and a panoramic radiograph of the lower limbs with blocks placed under the short leg until both iliac crests appear to be level were used for diagnosis. Doubtful cases were submitted to a CT scan. The radiography was used for planning based on the analysis of the Center-Edge angle (CEA), Acetabular Index (AI), and Neck-Shaft angle (NSA), which were measured at the first evaluation and at the last follow-up.

The aim of this procedure Is to convert the femur with delayed ossification (1b) to complete ossification (1a), and to correct the femoral deformity. Two cases in our cohort were treated differently because they ossified spontaneously, and the procedure was performed to improve their anatomy and biomechanics.

The operation was conducted according to Paley et al. [3] (Figure 1 and Figure 2). The procedure was divided in three parts: (1) Soft-tissue release (used to correct hip flexion, abduction, and external rotation, correcting contractures of the hip joint); (2) Femoral osteotomy (used to correct upper femoral varus, flexion, and external rotation, which corrects bony deformities of the femur); and (3) Pelvic osteotomy (used to correct a lack of femoral head coverage, resulting in better hip coverage).

After femoral osteotomy, the bone was fixated with different methods that varied according to the availability of specific materials for the surgery in our country. The following were used to fix the valgus osteotomy: a 130-degree fixed angle plate (3 patients), 120-degree pediatric LCP plate (20 patients), pediatric hip plate (1 patient), cannulated blade plate (1 patient), and Kirschner wires with ties (1 patient). Regarding acetabular osteotomies in young children used a Dega osteotomy. We needed in one case to perform a Ganz osteotomy in an older child.

### 2.3. Follow-Up

Clinical and radiographic assessments of the patients were performed during the pre- and post-operative evaluations.

*Clinical evaluation:* detailed physical examination, which included abduction in the supine position and internal and external hip rotations in the prone position (Figure 1).

Data was obtained from medical records and physical examinations were performed by the authors [3,9].

*Radiographic evaluation*: consisted of measuring CEA, AI, and NSA before and after surgery (Figure 3 and Figure 4). These assessments were performed by three different observers, including the senior author. A goniometer was used for the measurements [2,7,10,11].

### 2.4. Statistical Analysis

Descriptive statistics were created for all variables. A Pearson’s Chi-Square or Fisher’s exact test was used to test the association between categorical variables. The statistical significance of the differences in means between the quantitative variables was evaluated using the paired Student’s *t*-test and the existence of a correlation between the quantitative values was verified using Pearson’s correlation coefficient. The intraclass correlation coefficient (ICC) was used to check inter-observer reliability [12]. All results are reported according to the two-tail *p*-value calculation. All statistical analyses were performed using IBM SPSS Statistic 16.0 (IBM™ Corp, Armonk, NY, USA).

The data presented in this study are available upon request from the corresponding author. The data are not publicly available due to agreements with the patients in the ICF.

## 3. Results

This cohort include 26 patients (15 right and 11 left CFDs), with a mean age of 7 years (range 1 to 18 years). Most patients were female (17 patients; 65.4%). The lack of ossification in the neck was the most common radiographic finding (61.5%). Twenty (76.9%) patients were undergoing their first procedure and six (23.1%) had already undergone previous surgery.

Clinically, we assessed the range of motion, which included abduction, internal and external rotation, pre- and post-operatively (Table 1).

The angular measurements of the X-rays were analyzed by three experienced pediatric orthopedists before and after the surgeries. The reproducibility and reliability of the answers were confirmed by the ICC, all of which were classified as being in excellent agreement.

To compare the patient’s pre- and post-operative results, the average of the three observers was performed for each angle measurement. The average NSA angle was 72.3 ± 7.1° in the preoperative assessment and 133.1 ± 12.7° in the postoperative assessment, with an average increase of 60.8°, with statistical significance (*p* < 0.001). The preoperative measurement of the CEA showed a value of 16.8 ± 9.8° versus 33.5 ± 14.1° in the postoperative assessment, with an average increase of 16.7°, with statistical significance (*p* < 0.001). A statistically significant decrease was seen in the AI: 27.8 ± 6.9° in the first assessment and 16.4 ± 6.8° in the second assessment (*p* < 0.001).

All the data and results concerning the pre- and post-operative angular measurements were analyzed in relation to the age of the patients (two groups: those aged five years or less and those aged over 5 years), previous surgeries, gender, and type of material, as reported in Table 2. Examples are shown in Figure 5, Figure 6 and Figure 7.

There were no intra-operative complications, but during the post-operative period complications were found in four patients (15.4%). Of these, two did not ossify, one had wound dehiscence, and one had a broken plate. The rate of complications was compared in relation to age, previous surgery, gender, and type of implant. All the patients with complications were younger than 5 years old, which was considered a relevant factor for surgical complications (*p* = 0.011). The other variables were not statistically significant (Table 3).

## 4. Discussion

This is a large series of patients undergoing *SUPERhip* reconstruction surgery with Paley’s type 1b congenital femoral deficiency (CFD). This disease is a rare, complex malformation that is difficult for pediatric orthopedists to approach, and there have been few publications, except those by Paley et al., with details of the techniques used [1,3,4,5,7,8,9,11]. The number of patients in this study was 26; significant, considering the low incidence of CFD, and particularly of the specific subtypes Paley type 1a and 1b.

The average age was 7 years (1–18 years), which differs greatly from the classic indication for *SUPERhip*, which is between the ages of 18 months and 3 years. This disparity can be explained by the different socio-economic and educational conditions of the population and difficult access to the public health system, with few specialized sites that serve this population and a small number of experienced professionals in this specific area. Despite this, we were able to reproduce the objectives of *SUPERhip* surgery, i.e., improving hip stability and mobility and preparing patients for femoral lengthening surgery.

According to Herzenberg et al., femoral lengthening in congenital etiologies has a higher rate of complications, and the development of hip reconstruction surgeries is of crucial importance [13]. Hip congruence is especially important in this respect, as during lengthening, the force exerted by the adductor muscles generates a vector directed superolaterally through the hip joint, requiring proper planning of both femoral and acetabular procedures [2,14,15].

Clinical and radiographic monitoring of the hip is recommended to avoid subluxation, epiphysiolysis, and avascular necrosis [2,15]. The optimal management of subluxation during lengthening depends on degree of subluxation and the phase in which it occurs. Although instability most commonly occurs at the end of the distraction phase, patients can experience dislocation at any time [2].

Regarding acetabular containment, in older children, more complex osteotomies are an option, with procedures ranging from triple periacetabular (remain growth plates) to Ganz (in skeletally mature children) being performed to gain a greater and more three-dimensional reorientation of the acetabulum while sparing the growth plate. We needed to perform a Ganz osteotomy in one patient [7].

The evaluation of the patient’s pre- and post-operative results revealed an improvement in the average of all the radiographic parameters measured. The NSA represents the correction of femoral varus, with a clinical gain in abduction amplitude. The improvement of acetabular coverage is reflected by significant decreases in acetabular index measurements. These data are positive according to Bowen et al., who recommends the correction of femoral varus in hips with an NSA of less than 120 degrees and an AI greater than 25 degrees before femoral lengthening in patients with CFD [16]. Eidelman et al. has suggested surgical correction of proximal femoral varus (coxa vara) of less than 110 degrees before lengthening with valgus osteotomy. The CEA demonstrates the correction of acetabular dysplasia and hip joint congruence, varying according to the type of acetabuloplasty and the size of the bone graft introduced into the supra-acetabular region. Hips with a CEA greater than 20 degrees (33 degrees in average in our study) during lengthening tend to have lower complication rates [17].

Studies reporting the outcome of pre- and post-operative clinical assessments of joints’ range of motion are rare. In most cases of CFD, internal rotation is blocked at neutral and an external rotation at 90 degrees. This is different from what was found in our sample, which found a significant increase in internal rotation post-operatively (31.2 ± 17.1°), correcting the femoral retroversion. Considering that an abduction arc of less than 30° during lengthening is a risk for dislocation [2,14,16], this gain of 12 degrees of abduction protects the patients in future lengthening procedures.

The type of plate most used was a locking compression pediatric hip plate (LCP), with two different versions, and in two patients Kirschner wires were used (Wagner configuration); there was no recurrence of the varus deformity. However, two patients had non-unions, and one of them had an implant failure. The use of the LCP-type plate provided a significant improvement in the results of the correction of the radiographic values measured, even though it was not related to the correction of CEA and AI angle values, which depend on the quality of the acetabular osteotomy and the size of the bone graft.

As a limitation, this is a retrospective cohort study subject to bias associated with the study design. The CFD sample size is small, justified by the rarity of this malformation. A prospective study involving a larger number of patients is needed in the future to assess the validity of this novel surgical procedure. However, the minimal and mean follow-up of our patients were of a sufficient length to provide medium-term data, and no patient was lost to follow-up. For future research, a longer follow-up should be performed to verify the incidence of late complications, such as the recurrence of varus, as described by Paley.

## 5. Conclusions

The *SUPERhip* procedure, for the treatment of congenital femoral deficiencies, proved to be effective for the clinical and radiographic correction of this condition to a significant degree. In addition, there was excellent agreement in the evaluation of the angular means between the evaluators. There was no evidence of an influence of age, gender, or previous surgeries, although the LCP-type plate had better results, and being aged below 5 years was a relevant factor for complications.

This technique is reproducible and will allow for safer femoral lengthening in this complex condition. It should replace other, not-so-efficient procedures.

## Figures and Tables

**Figure 1 children-11-00075-f001:**
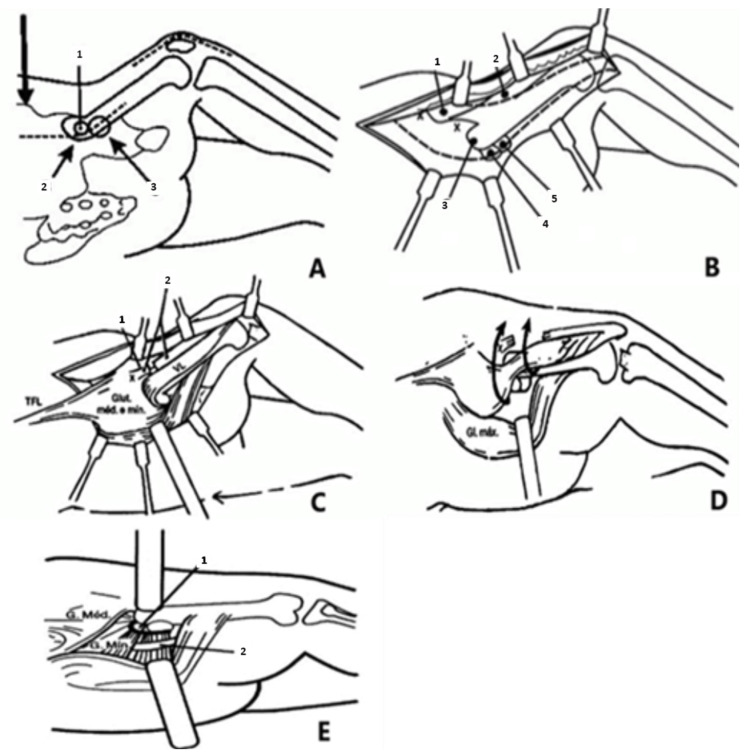
*SUPERhip* Soft Tissue Release. (**A**) Supine positioning—Incision in posterior aspect of iliac crest, over the greater trochanter and then in-line with the femoral diaphysis. (1—Greater trochanter; 2—Lateral “bump” of the diaphysis; 3—Femoral head). (**B**) Fascia and tensor fascia lata reflection. (1—Lateral border of sartorius; 2—Femoral quadriceps; 3—Lateral “bump” of the diaphysis; 4—Greater trochanter; 5—Femoral head). (**C**) Anterior release of the hip (1—Sartorius muscle; 2—Rectus femoris tendon—split; TFL—Tensor Fascia Lata; VL—Lateral vastus). (**D**) Elevation of the continuous flap with the gluteal muscles (medius and minimus) and the vastus lateralis to the anterior. (**E**) Release of abduction contracture—split of piriform tendon (1—Piriform tendon; 2—Sciatic nerve). Redrawn from section written by Dror Paley in the book Limb Lengthening and Reconstruction Surgery, edited by S. Robert Rozbruch [5].

**Figure 2 children-11-00075-f002:**
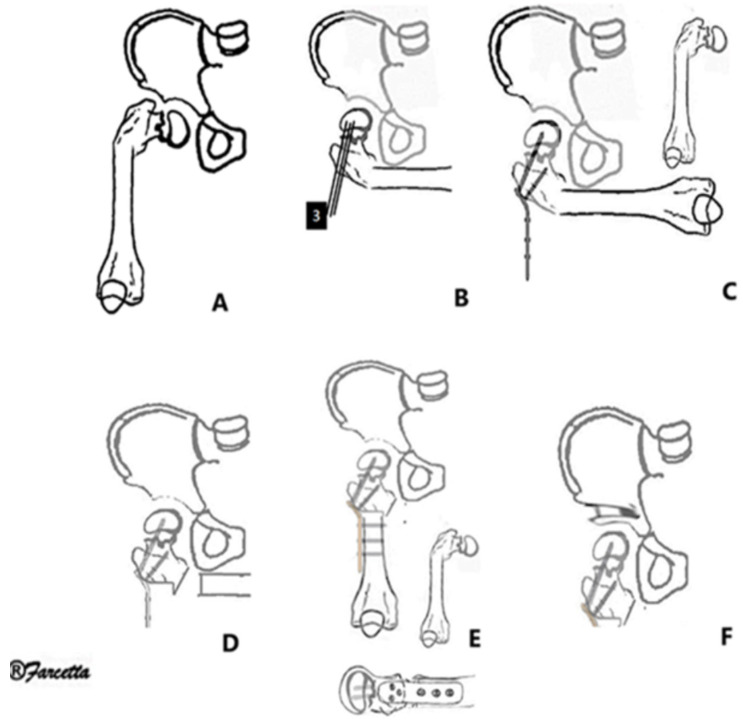
*SUPERhip* femoral and pelvic osteotomies. (**A**) Schematic design of a femoral deformity in type 1b Paley CFD. (**B**) To better visualize the proximal segment, the femur is placed in maximum adduction, flexion, and external rotation. First pin is inserted to guide the angulation of the neck (retroversion). The other two pins are inserted using the plate as a guide, total of three guide wire pins (proximal fixation of the plate). (**C**) Locked plate is fixed by proximal screws in lateral view. (**D**) An osteotomy is performed, preserving part of the medial cortex for anchorage and lateralization of the diaphysis. In CFD, shortening is necessary to ensure less tension. (**E**) The rest of the plate is fixed without tensioning the soft tissue (Correction in AP and lateral view). (**F**) In addition, there must be adequate pelvic coverage; this example is a Dega osteotomy.

**Figure 3 children-11-00075-f003:**
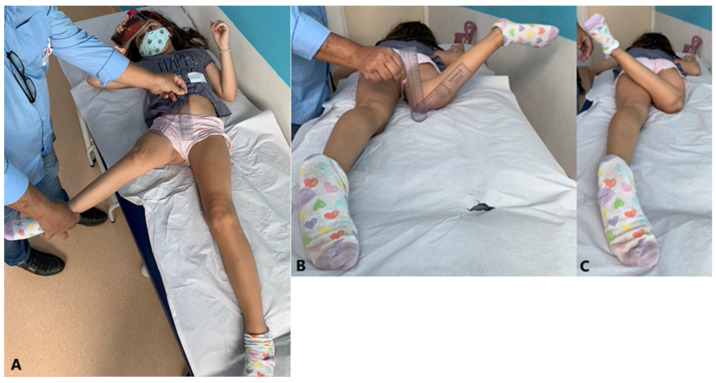
Clinical assessment. (**A**) Abduction measurement. (**B**,**C**), respectively, show internal and external rotation measurements.

**Figure 4 children-11-00075-f004:**
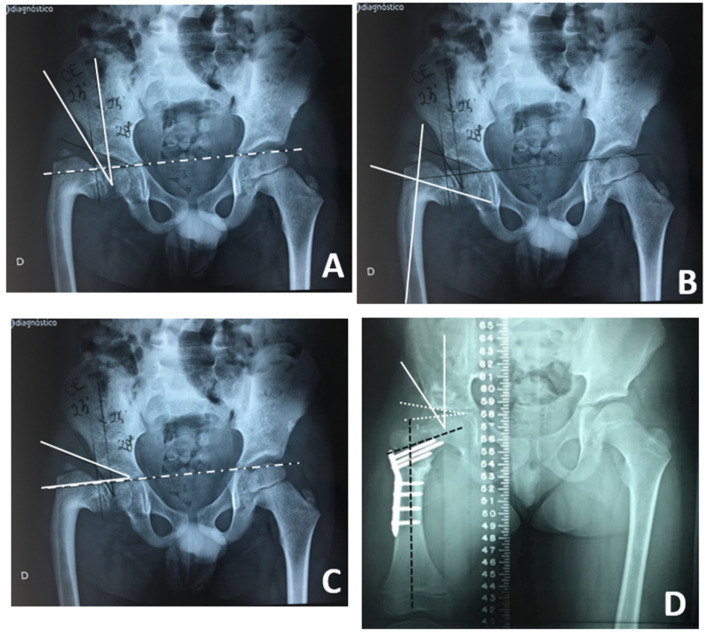
Pre-operatory planning: (**A**) Center-Edge angle (CEA). (**B**) Neck-Shaft Ankle (NSA). (**C**) Acetabular Index (AI). Post-operatory: (**D**) All measurements corrected.

**Figure 5 children-11-00075-f005:**
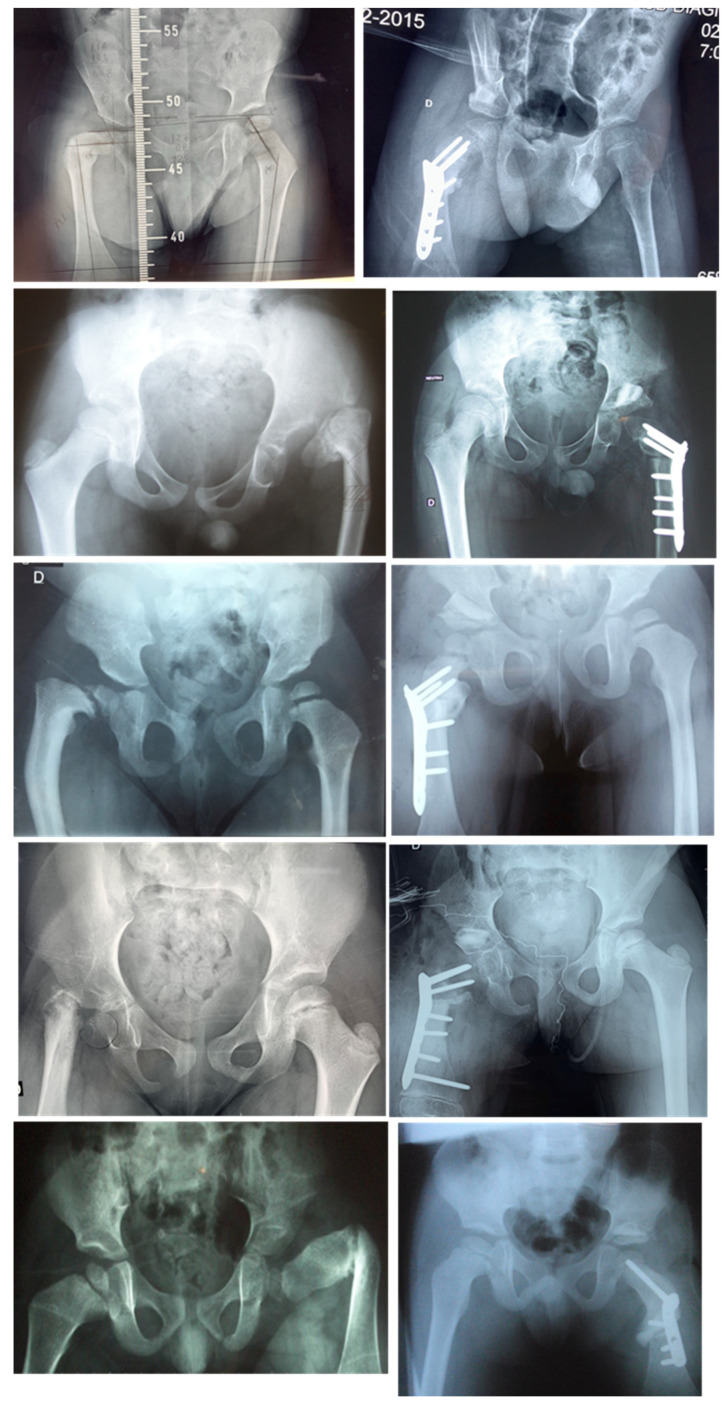
Pre- and Post-operation for delayed ossification in the femoral neck, treated with LCP.

**Figure 6 children-11-00075-f006:**
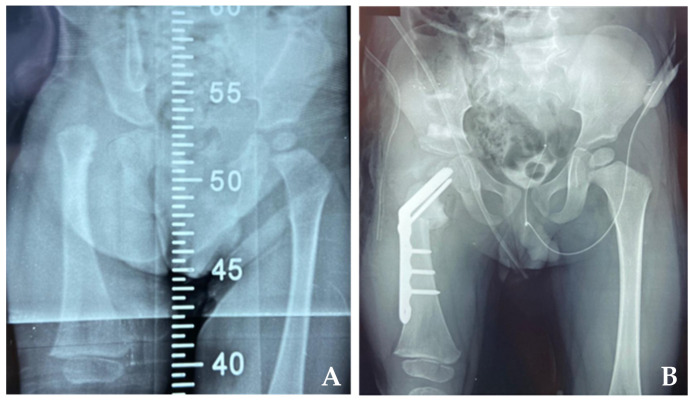
Patient treated with a blade plate. (**A**) Pre-operatory radiography AP with delayed ossification of head and neck. (**B**) After surgery, in which the anatomy of the proximal femur were restored.

**Figure 7 children-11-00075-f007:**
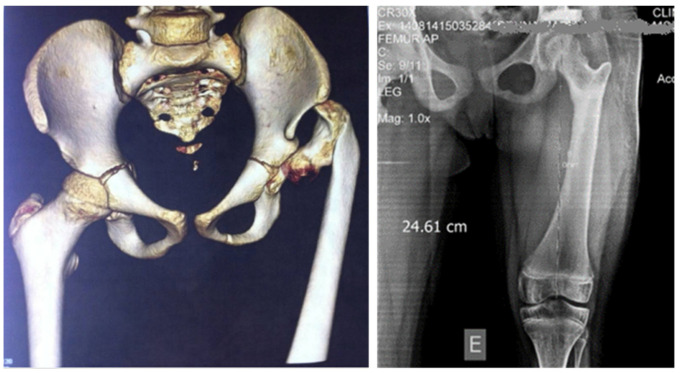
Case with 5 years post-operatory radiography AP of femur (right image), with restored anatomy of the femur, prepared for lengthening.

**Table 1 children-11-00075-t001:** Comparison of range of motion, pre- and pos-operation.

Range of Motion	Pre-Operatory Mean ± SP	Post-Operatory Mean ± SP	Diference	*p*-Value
Abduction	32.1 ± 9.3	44.3 ± 13.3	12.2	<0.001
Internal Rotation	3.0 ± 10.2	31.2 ± 17.1	28.2	<0.001
External Rotation	80.8 ± 10.0	51.3 ± 17.4	−29.5	<0.001

**Table 2 children-11-00075-t002:** Analysis of measured angles according to different variables: gender, age, previous surgery, and type of implant.

Angulation	Pre-Op Mean ± SD	Post-Op Mean ± SD	Comparison		
			*p* ^1^	*p* ^2^	*p* ^3^
**NSA**					
Female	72.4 ± 28.1	133.9 ± 13.5	<0.001	0.625	0.691
Male	72.0 ± 26.8	131.7 ± 11.6	<0.001		
< or = 5 years	73.7 ± 28.8	133.9 ± 9.8	<0.001	0.31	0.790
>5 years	71.2 ± 26.7	132.6 ± 14.8	<0.001		
With previous surgery	77.7 ± 26.2	128.7 ± 12.3	0.002	0.762	0.341
Without previous surgery	70.6 ± 27.8	134.5 ± 12.8	<0.001		
LCP	72.1 ± 26.0	132.1 ± 13.2	<0.001	0.791	0.449
Other plates	72.9 ± 33.3	136.7 ± 11.2	0.001		
**CEA**					
Female	14.5 ± 9.7	31.2 ± 9.1	<0.001	0.106	0.247
Male	21.0 ± 8.9	38.0 ± 20.5	0.037		
< or = 5 years	17.6 ± 7.5	32.5 ± 6.9	<0.001	0.72	0.761
>5 years	16.2 ± 11.4	34.3 ± 17.9	0.003		
With previous surgery	11.2 ± 11.0	31.2 ± 10.6	0.009	0.112	0.650
Without previous surgery	18.4 ± 9.0	34.2 ± 15.2	0.003		
LCP	14.2 ± 9.5	31.4 ± 8.3	<0.001	0.013	0.161
Other plates	25.2 ± 5.3	40.7 ± 25.6	0.177		
**IA**					
Female	27.7 ± 5.3	15.2 ± 7.5	<0.001	0.258	0.200
Male	30.0 ± 9.3	18.8 ± 4.6	0.005		
< or = 5 years	25.2 ± 4.6	16.3 ± 6.3	0.001	0.104	0.962
>5 years	29.7 ± 7.8	16.5 ± 7.3	<0.001		
With previous surgery	27.3 ± 6.7	16.9 ± 4.7	0.01	0.845	0.848
Without previous surgery	28.0 ± 7.2	16.3 ± 7.4	<0.001		
LCP	28.7 ± 6.4	15.9 ± 7.2	<0.001	0.282	0.523
Other plates	25.1 ± 8.6	18.0 ± 5.4	0.117		

*p* ^1^ is the comparison between Pre-op and Post-op for each line; *p*
^2^ is the comparison between Pre-ops for each variables (age, gender, previous surgery and type of implant); *p*
^3^ is the comparison between Pos-ops for each variables (age, gender, previous surgery and type of implant).

**Table 3 children-11-00075-t003:** Complication comparative analysis.

Variable	Complications		
	Yes	No	*p*-Value
	*n* (%)	*n* (%)	
Age			
≤5 years	4 (100.0)	7 (31.8)	0.011
>5 years	0 (0.0)	15 (68.2)	
Previous surgery			
Yes	0 (0.0)	6 (27.3)	0.234
No	4 (100.0)	16 (72.7)	
Gender			
Female	3 (75.0)	14 (63.6)	0.66
Male	1 (25.0)	8 (36.4)	
Type of implant			
LCP	3 (75.0)	17 (77.3)	0.921
Others	1 (25.0)	5 (22.7)	

## Data Availability

Data are contained within the article.

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
