# Peer review of "SUPERhip Reconstruction Treatment in Patients with Congenital Femoral Deficiency of Paley’s Classification Type 1b"

_children, 2024, doi:10.3390/children11010075_

Round 1
Reviewer 1 Report
Comments and Suggestions for Authors
Introduction:
Specify the target audience for the article (e.g., orthopedic surgeons, researchers, medical professionals) to provide context for the level of detail and technical language used.
Clarify the significance of Paley's classification and its relevance to the SUPERhip procedure.
Materials and Methods:
Provide a brief overview of the hip procedure at the beginning of the section to give readers a clear understanding before delving into the details.
Define abbreviations, such as CFD, NSA, CEA, AI, etc., upon first use for better reader comprehension.
Consider breaking down complex sentences for improved readability.
Results:
Use concise and clear language to present results, focusing on key findings.
Consider presenting statistical information in a more reader-friendly format, such as tables or graphs.
Clarify whether the complications mentioned occurred during or after the hip procedure.
Discussion:
Provide a more explicit connection between the study's results and existing literature.
Discuss limitations of the study in more detail and propose areas for future research.
Highlight the clinical implications and potential impact of the hip procedure on patient outcomes.
Conclusion:
Recap the key findings succinctly and emphasize their clinical relevance.
Clearly state the practical implications and potential benefits of the hip procedure.
General Suggestions:
Ensure consistency in formatting, particularly in the citation style.
Double-check grammar and punctuation for accuracy.
Author Response
Dear reviewers,
Thank you for your comments.
All the reviewers' points have been answered in the text of the resubmitted article.
Sincerely,
Monica Paschoal Nogueira
Reviewer 2 Report
Comments and Suggestions for Authors
The article presents intriguing insights into a complex pathology and its intricate surgical treatment. It is well-written and structured.
My only suggestion is to revise the statistical method, recommending the inclusion of multivariate analyses and possible linear or logistic regressions to assess the impact of age and other variables on correction. A Mann-Whitney test could be useful to examine age differences between patients with and without complications.
I'm also interested in knowing how many patients underwent subsequent limb lengthening.
Thank you for your anticipated responses.
Comments on the Quality of English Languageenglish is fine to me.
Author Response

(The authors gave the same response as above.)
